# Design and Simulation of Terahertz Perfect Absorber with Tunable Absorption Characteristic Using Fractal-Shaped Graphene Layers

**Amir Maghoul [1,*], Ali Rostami [2], Nilojan Gnanakulasekaran [3] and Ilangko Balasingham [1,3]**

1   Department of Electronic Systems, Norwegian University of Science and Technology,
    7491 Trondheim, Norway; ilangko.balasingham@ntnu.no
2   Photonics and Nanocrystal Research Lab (PNRL), Faculty of Electrical and Computer Engineering,
    University of Tabriz, Tabriz 5166614761, Iran; rostami@tabrizu.ac.ir
3   Intervention Center, Oslo University Hospital, 0027 Oslo, Norway; nilojan.gnanakulasekaran@ntnu.no
*   Correspondence: amir.maghoul@ntnu.no

**Abstract:** Graphene material, due to its unique conductivity and transparency properties, is utilized extensively in designing tunable terahertz perfect absorbers. This paper proposes a framework to design a tunable terahertz perfect absorber based on fractal triangle-shaped graphene layers embedded into dielectric substrates with the potential for spectral narrowing and widening of the absorption response without the need for geometric manipulation. In this way, the absorption cross-section spectra of the suggested configurations are achieved over the absorption band. First, the defection impact on the single-layer fractal triangle-shaped graphene structure inserted in insulators of the absorber is evaluated. Then, a flexible tunability of the absorbance's peak is indicated by controlling the Fermi energy. By stacking fractal graphene sheets as a double graphene layer configuration in both the same and cross-states positioning, it is demonstrated that the absorption characteristics can be switched at 6–8 THz with a stronger amplitude, and 16–18 THz with a lower intensity. The impact of changing the Fermi potentials of embedded graphene layers is yielded, resulting in a plasmonic resonance shift and a significant broadening of the absorption bandwidth of up to five folds. Following, the absorption spectra related to the fractal triangle-shaped structures consist of a multi-stage architecture characterized by a spectral response experiencing a multiband absorbance rate and an absorption intensity of over $8 \times 10^6$ nm$^2$ in a five-stage perfect absorber. Ultimately, the variations of the absorbance parameter and plasmonic mode under rotating the graphene sheet are explored for single and double fractal triangle-shaped perfect configurations on the absorption band. The presented mechanism demonstrates the tunability of the absorption spectrum in terms of narrowing or broadening and switching the plasmonic resonance by configuring multi-stage structures that can employ a broad range of applications for sensory devices.

**Keywords:** tunable; fractal graphene layer; perfect absorber; multi-stage structure; plasmonic mode; spectral narrowing/widening





## 1. Introduction

The emergence of the terahertz regime's high-tech technologies has developed microelectronics knowledge and low-dimensional devices, such as filters, detectors, and absorbers, in the different fields of applied science and the industry [1–6]. Among these, absorbers [7–10] have attracted particular attention due to their optical performance in a wide range of applications, including spectroscopy [11], biosensing [12,13], optical control [14,15], and solar cells [16–18]. In this context, using conductive structures with sub-wavelength dimensions has the potential to generate localized surface plasmon resonances (LSPR) on the absorption band, which are dependent on the opto-geometric characteristics and physical properties of their materials [19]. In further detail, the geometrical structure

and size of the conductive layers play a decisive role in creating plasmonic resonance in the terahertz absorbers' spectral response. In attempting to realize tunable multiband or broadband terahertz perfect absorbers, many different structures have also been suggested that use multiplexed planar, which involves stacking multiple layers separated by the dielectric layer [20]. In between, graphene is the most widely applicable material used to design perfect absorbers as a conductive layer [21]. Graphene is a 2D material composed of carbon atoms arranged in a hexagonal honeycomb lattice. This material has a unique and distinguishable performance compared to other 2D materials due to its optical, electrical, chemical, and mechanical properties [22]. From an optical point of view, the dependency of conductivity to the frequency in graphene layers acts as an important factor in designing terahertz components and many other high-frequency applications, which consist of two types: intraband and interband conductivity. Besides, graphene layers are designed for different geometrical configurations that operate as frequency selective surfaces (FSSs) [23]. For instance, using fractal configurations [24], ring structures [25] in constructing FSS layers as an ingredient of metamaterial structures are very common in the terahertz regime. It is also necessary to point out that tuning surface plasmon polariton on the absorption band for absorber devices that used metal surfaces in their construction is very difficult, due to a high sensitivity to polarization and incident angles. In return, utilizing graphene layers might be beneficial because of polarization insensitivity, multi-angle incidence, and high tunability [26]. Recently, some works have been reported in which novel perfect multi-band absorbers based on the different shaped graphene layers [27,28], such as the T-shaped structure [29,30], were introduced.

Moreover, due to the role of fractal theory in a wide range of applications, considering this concept in the design of optoelectronic devices leads to high-sensitive absorbers based on the fractal configurations in emerging THz technologies appearing. Fractal structures have the advantages of miniaturization, multiple resonances, the multiscale nature of their architecture, and a considerable absorbance [31]. Compared to the periodic structures that generate resonant frequencies corresponding to $f_0$, $2f_0$, $3f_0$, ... , the fractal structures give the possibility for the photonic devices to operate with the specific resonant frequencies, rather than multiple basic frequencies in general over the absorption band. Although some studies concerning the design of fractal absorbers have been carried out in the THz regime recently [32–34], there is still substantial potential for further research in this field. Accordingly, this article proposes a unique terahertz perfect absorber composed of the graphene layer, configuring triangular fractal shapes, two insulators, and a thickness substrate of gold. First, the effect of an increase in the number of fractal triangles shaped on the graphene sheet embedded in the proposed configuration is investigated in order to obtain the absorption parameter changes in the spectral response. Then, the impact of stacking fractal layers of graphene, position variations of fractal layers on the plasmonic resonance shift, and absorption peak amplitude in the absorption characteristic are studied. Further, interesting results are extracted using the DC voltage bias employed by external sources on the configuration of the proposed multilayer absorbers, which means tuning the amplitude and frequency of the absorption and narrowing or widening the absorption bandwidth of the spectral response and multiband absorption.

## 2. Theory and Background

In general, an absorber is a strong collector of electromagnetic waves that can trap incident waves in specific locations on the spectral region. From a practical perspective, absorbed waves can be turned into different forms of energy, ranging from microwave to the optical regime [35]. Thin conductive surfaces with a specific conductivity at the flat boundary between two dielectrics with different refractive indexes function similarly to a load attached to the junction of two transmission lines [36]. In absorbers that consist of graphene, the graphene's conductivity surface plays the main role in determining the

absorption band's resonant peak. Graphene's conductivity surface can be described by the Kubo formula [37], which includes interband and intraband parts, as:

$$\sigma\left(\omega, E_f, \Gamma, T\right) = \sigma_{int\ er} + \sigma_{int\ ra,} \tag{1}$$

$$\sigma_{int\ ra} = \frac{2k_B T e^2}{\pi \hbar^2} \ln\left(2cosh\frac{E_f}{2k_B T}\right)\frac{i}{(\omega + i\Gamma)} = \frac{\alpha}{-i\omega + \Gamma} \tag{2}$$

$$\sigma_{int\ er} = \frac{e^2}{4\hbar}\left[H\left(\frac{\omega}{2}\right) + i\frac{4\omega}{\pi}\int_0^\infty \frac{H(\Omega) - H\left(\frac{\omega}{2}\right)}{\omega^2 - 4\Omega^2}d\Omega\right] \tag{3}$$

$$H(\Omega) = \frac{sinh\left(\frac{\hbar\Omega}{k_B T}\right)}{\left[cosh\left(\frac{\hbar\Omega}{k_B T}\right) + cosh\left(\frac{E_f}{k_B T}\right)\right]} \tag{4}$$

In the above formulas, $\omega$ is the angular frequency, $E_f$ describes graphene's Fermi energy, $T$ is the temperature, $k_B$, and $e$ is the Boltzmann constant and elementary charge. In addition, $\hbar$ expresses the reduced Planck constant. Typically, if the Fermi energy becomes bigger compared to the photon energy ($\hbar\omega/2$), the interband section of Equation (1) becomes negligible compared to the intraband section, due to Pauli blocking. The intraband part of graphene's conductivity is approximated by the Drude model, as demonstrated in Equation (2). In addition, the effective dielectric constant can be calculated by:

$$\varepsilon = 1 + i\sigma/\varepsilon_0 \omega t_g \tag{5}$$

Accordingly, the relationship between the plasma frequency of graphene and graphene's thickness is obtained as follows:

$$\omega_p = \left[\frac{2e^2 k_B T}{\pi \hbar^2 \varepsilon_0 t_g}\ln\left(2cosh\frac{E_f}{2k_B T}\right)\right]^{1/2} = \sqrt{\frac{\alpha}{\varepsilon_0 t_g}} \tag{6}$$

On the other hand, one of the most applicable methods used for shifting absorption peaks is to apply an external voltage bias and stimulate the graphene layer by changing the Fermi level. Overall, the relationship between the Fermi energy, $E_f$ of the graphene layer and voltage bias $V_{DC}$ can be calculated by:

$$\left|E_f\right| = \hbar v_f \sqrt{\frac{\pi \varepsilon_r \varepsilon_0 V_{DC}}{e t_s}} \tag{7}$$

where $\varepsilon_0$ and $\varepsilon_r$ are the permittivity of the vacuum and spacer, respectively. In addition, $t_s$ and $V_{DC}$ express the spacer's thickness and *DC* voltage bias. Further, $v_f$ named the Fermi velocity as equal to $1.1 \times 10^6$ (m/s). The required external voltage for changing the Fermi potential can be approximated from Equation (7).

## 3. Structural Geometry

The schematic demonstration of the proposed structure, which configures two insulators and a graphene layer between them placed on a thick gold substrate, is illustrated in Figure 1a. The graphene layer's triangular shapes are designed based on the fractal established concepts, as shown in Figure 1b. The primary geometrical size of the absorber configuration is addressed in Table 1. The thickness of the graphene layer is sized as 1 nm, and the two insulators' permittivity is equal to 1.96. As such, the optical properties of gold are defined by the Drude model. On this basis, the bulk gold's permittivity can be described by $\varepsilon_\infty = 1$, the plasma frequency $\omega_p = 1.37 \times 10^{16}$ s$^{-1}$, and damping constant $\omega_r = 1.23 \times 10^{14}$ s$^{-1}$ [38]. The conductive surface of the graphene layer is attributed for full-wave electromagnetic analysis and simulation. Further, we set a temperature of 300 K and a relaxation time of 1 ps in order to stimulate the graphene's conductive layer.

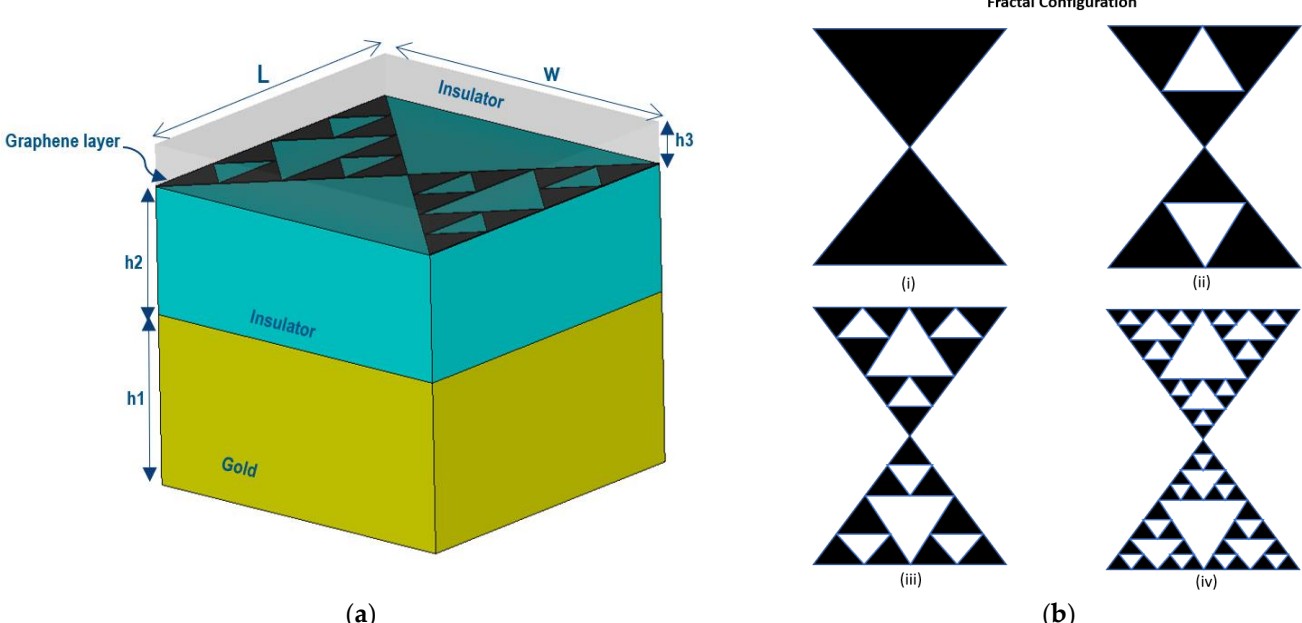

**Figure 1.** (**a**) The schematic diagram of perfect absorber based on the fractal graphene layer; (**b**) triangular fractal shapes on the graphene layer.

**Table 1.** The initial geometrical size of the absorber structure.

| h1 | h2 | h3 | L | W |
|---|---|---|---|---|
| 3 μm | 1.5 μm | 50 nm | 0.4 μm | 0.4 μm |

The simulations are performed using numerical methods based on the finite integration technique (FIT) in CST Microwave Studio. A time-domain solver is employed to extract absorption spectra related to the developed perfect absorber.

## 4. Simulation and Discussion

At the start, the proposed absorber architecture is implemented in the software environment based on the sizes available in Table 1 and the fractal shapes of Figure 1b. A plane wave with an electric field intensity of 1 v/m as the excitation wave is incident on the configuration from the top. In other words, the incident vector is perpendicular to the surface in which the fractal graphene layer is available. Due to the great thickness of the gold substrate, the transmittance coefficient of this configuration is negligible. Indeed, the gold structure has a mirror function against excitation wave irradiation, which means that the absorbance and reflectance count as the two main parameters in the configuration's optical characterization. First, we use a fractal layer similar to a triangle bow shape, as illustrated in Figure 1(bi). As such, we consider a Fermi level ($E_f$) of 0.7 eV for a triangular-bow-shaped graphene layer at the beginning of the simulation. While the Fermi level is fixed at 0.7 eV, fractal architectures with shapes ii–iv (shown in Figure 1) are employed. In order to evaluate the proposed structures in terms of functionality, the absorption cross-section (ACS) parameter as a measure of the absorption process is obtained, and its relationship with the absorption efficiency is defined as follows [39]:

$$Q_{abs} = \sigma_{abs} / A \tag{8}$$

where $\sigma_{abs}$, $Q_{abs}$, and $A$ are the absorption cross-section (ACS) normalized, absorption efficiency, and cross-sectional area, and the absorption efficiency corresponds to the cross-section normalized to the geometrical area of the structure.

The simulated results related to the absorption cross-section (ACS) characteristic of the presented structures in Figure 2a illustrate the amplitude of the resonance peaks arising from the plasmonic mode on the absorption band. Creating the fractal triangle shapes on the graphene layer reduces the graphene layer's effective radiation surface, which causes a significant drop in the peak of absorption intensity, as illustrated in Figure 2a. By contrasting the simulation results arising from the variations of the fractal graphene layer based on Figure 1(biii,biv), it can be comprehended that an increase in the number of fractal triangle defects on the graphene layer changes the absorption characteristic of the absorber. As can also be seen, when the plasmonic resonant frequency shifts, the magnitude of ACS dramatically changes. In detail, the digging of fractal triangles over the graphene layer leads the absorption area to decrease, consequently reducing the absorption peak due to diminishing the electron density upon the graphene layer. As such, continuing this process, such as in Figure 1(biv), leads to a saturation in the absorption spectrum. This behavior may result from the reflected wave's phase mismatching in the absorption band. In the following, we choose a developed absorber configuration with a fractal triangle graphene layer similar to Figure 1(biii) to investigate its optical behavior in the absorption band. Figure 2b shows the electromagnetic fields' distribution in the TE and TM modes and the surface current at a plasmonic frequency of 5.2285 THz related to the selected fractal structure under a Fermi energy of 0.7 eV. As can be seen, the effect of the electric field intensity is more considerable in the graphene layer's edges, whereas the magnetic field intensity is a predominant factor in the connection of two triangle fractal surfaces.

In general, the dimension change of the insulators and their optical properties, such as their refractive index, can directly influence the absorption characteristic of perfect absorbers, which is comprehensively reported in [40,41], and their impacts on the absorbance rate have been investigated, so we neglect to study them more here in this field. Apart from them, it has also been proven that graphene's Fermi energy can be tuned using external DC voltage [42]. For investigation of this feature in the suggested structure, the Fermi level of the selected fractal structure is manipulated between 0.3 eV and 1.5 eV (this is only for simulation, and is not practical in the actual model). By increasing the Fermi potential, the peak amplitude can be changed almost between $1.011 \times 10^5$ nm$^2$ at $E_f = 0.3$ eV, $2.5998 \times 10^5$ nm$^2$ at $E_f = 0.9$ eV, and $1.785 \times 10^5$ nm$^2$ at $E_f = 1.5$ eV. The plasmonic frequency also shifts from 3.47 to 5.932 THz and 7.6513 THz, respectively, without any geometric manipulation. It is interesting to note that, after applying 1.4 eV of the Fermi potential, the spectral response is widened with a dramatic drop in the absorption peak, which can be due to the nature of this kind of fractal shape, and we focus further on this issue in the following. In fact, changing the Fermi energy leads to altering the graphene's optical properties, such as its refractive index and permittivity, resulting in the variation of plasmonic frequency on the spectral response. In detail, the resonant frequency of the proposed absorber structure can be approximately expressed as $\omega = 1/\sqrt{Lc}$, in which L and C are the inductance and capacitance, respectively. The total inductance is equal to the summation of the kinetic inductance ($L_k$) and standard inductance ($L_g$). In [43], the kinetic inductance ($L_k$) is described as $L_k = \alpha(m_e/(N_d e)^2)$, where $\alpha$ is related to the unit cell's structure, and $e$ and $m_e$ are the electron charge and mass. $N_d$ also shows the carrier concentration. By increasing the graphene's Fermi level, the kinetic inductance is reduced; consequently, the total inductance decreases. Therefore, the resonant frequency increases, which are associated with the absorption amplitude amplification, move in an upward trend, as shown in Figure 3 [44]. Table 2 presents the quantitive variation of plasmonic resonance associated with the corresponding absorption amplitude under the impact of Fermi level changes in the single-layer fractal absorber proposed.

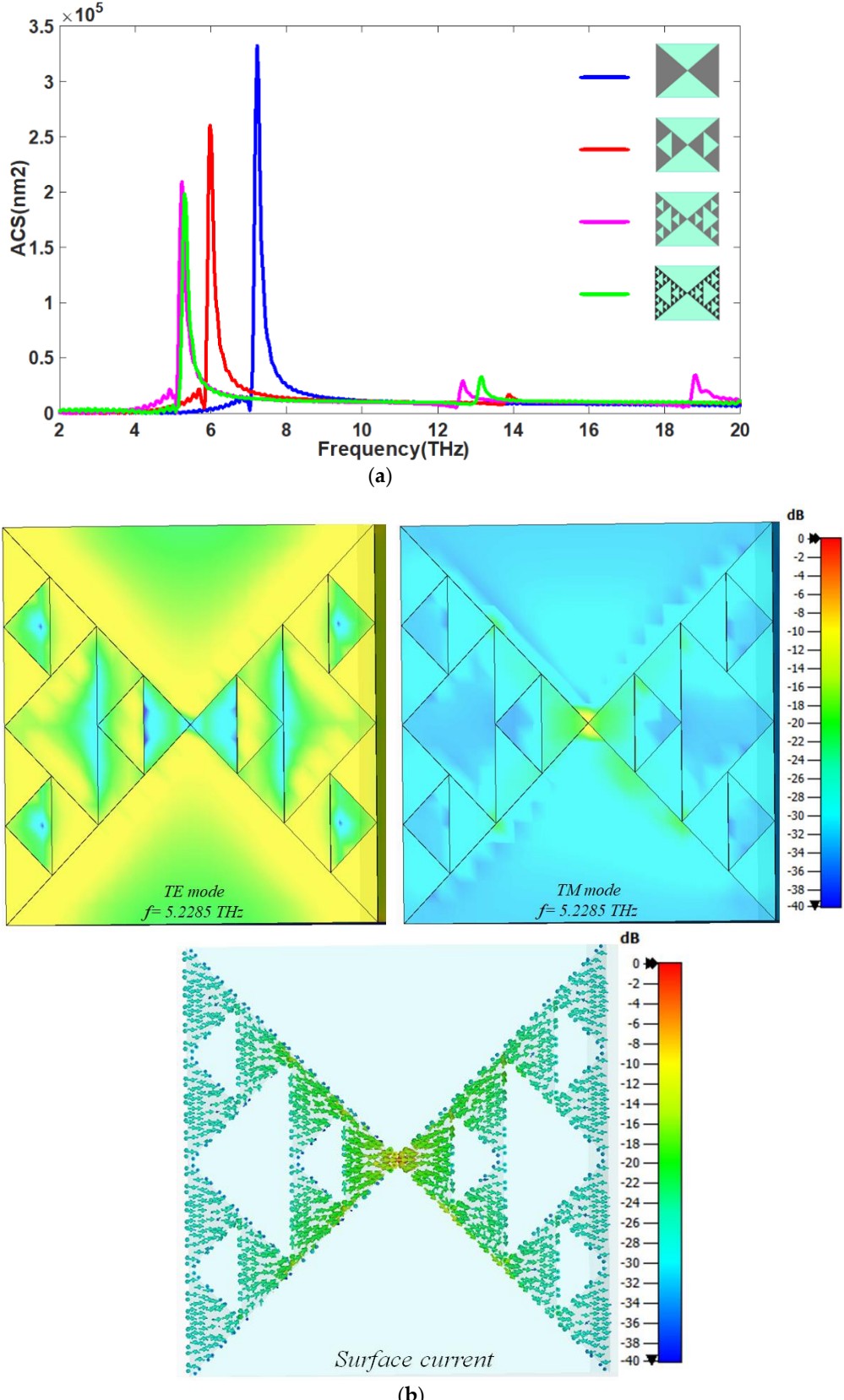

**Figure 2.** (**a**) Absorption cross−section (ACS) spectra for different fractal triangle−shaped graphene layers embedded into perfect absorbers; (**b**) the TE and TM electromagnetic field distribution and surface current over the fractal triangle−shaped graphene layer selected in its plasmonic resonant frequency.

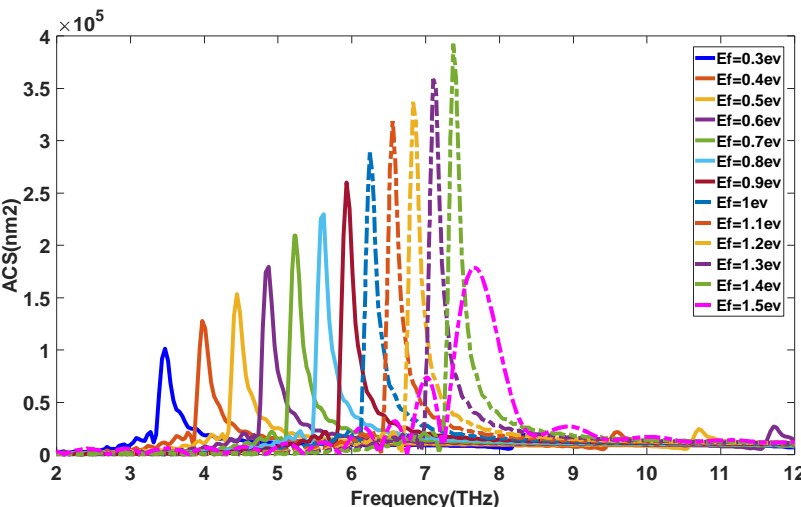

**Figure 3.** The variation of ACS spectrum under changing the Fermi energy in a single fractal graphene layer.

**Table 2.** The variations of plasmonic resonance and absorption peak under Fermi level change for a single fractal graphene layer.

| Fermi Potential (eV) | 0.3 | 0.4 | 0.5 | 0.6 | 0.7 | 0.8 | 0.9 | 1 | 1.1 | 1.2 |
|---|---|---|---|---|---|---|---|---|---|---|
| Plasmonic resonant frequency (THz) | 3.47 | 3.978 | 4.447 | 4.8768 | 5.2285 | 5.6192 | 5.932 | 6.2445 | 6.5571 | 6.8307 |
| Amplitude of absorption peak ($\times 10^5$ nm$^2$) | 1.011 | 1.2775 | 1.529 | 1.7922 | 2.0892 | 2.2945 | 2.5998 | 2.898 | 3.186 | 3.3675 |

Next, one fractal graphene sheet associated with a similar insulator to the previous one is stacked on the initial configuration, as demonstrated in Figure 4a. In other words, the fractal graphene's layers are embedded in the same insulators stacked with the same Fermi level of 0.7 eV in the two-stage structure as a double fractal absorber. As before, the plane wave is incident on the structure from the top. The achieved spectral response of ACS in Figure 4b illustrates the absorption peak's amplitude of approximately $3.7326 \times 10^5$ nm$^2$ at 6.5962 THz, whereas that of the initial structure was equal to $2.0892 \times 10^5$ nm$^2$ at 5.2285 THz. Adding another graphene layer to the initial structure leads to enhancing the plasmonic modes, resulting in a sharper absorption response due to the constructive interference of the electromagnetic field components. The absorption bandwidth in the new configuration also slightly increases. These phenomena can be described by the circuit theory, in which the graphene sheet is modeled by a shunt admittance [45]. Following this, the equivalent circuit of the configuration is approximated by the transmission line and graphene admittance. Based on this established concept, a near-perfect absorption occurs at the specific frequency when the input admittance of the structure matches with the free space admittance. Hence, in order to realize a broadband spectrum, the designers need to achieve plasmonic peaks close to each other merged under the admittance matching over the absorption band, resulting in a broadening absorption response [46]. Founded by this, the same graphene layers are used to obtain stronger plasmonic resonances. In our proposed model, due to the admittance matching close to each other related to the graphene sheets, the amplitude of the absorption peak experiences a considerable amplification, as seen in Figure 4c.

In the following section, the lower graphene layer used for the two-stage structure is rotated as much as 90°, such as in Figure 4b, which is referred to as a cross-state of the graphene layers inside the proposed absorber. The rotation of the lower graphene sheet alters the reflected electromagnetic polarization; correspondingly, the electromagnetic field components related to the absorbance coefficient are also varied, which means the changing of the plasmonic resonance on the absorption band. As mentioned in Table 3, two peaks

on the spectral response are observed at 4.52 THz with an amplitude of $6.6911 \times 10^4$ nm$^2$, and 16.285 THz with an amplitude of $18.7564 \times 10^4$ nm$^2$, under this rotation, resulting from admittance mismatching, which creates a specific distance between the plasmonic peaks that occur in this state. Figure 5 shows the profile of the electromagnetic fields for the TE and TM modes related to the plasmonic resonance in both configurations, in which variations of electromagnetic modes under the rotation of the graphene layer in plasmonic frequencies are well observed.

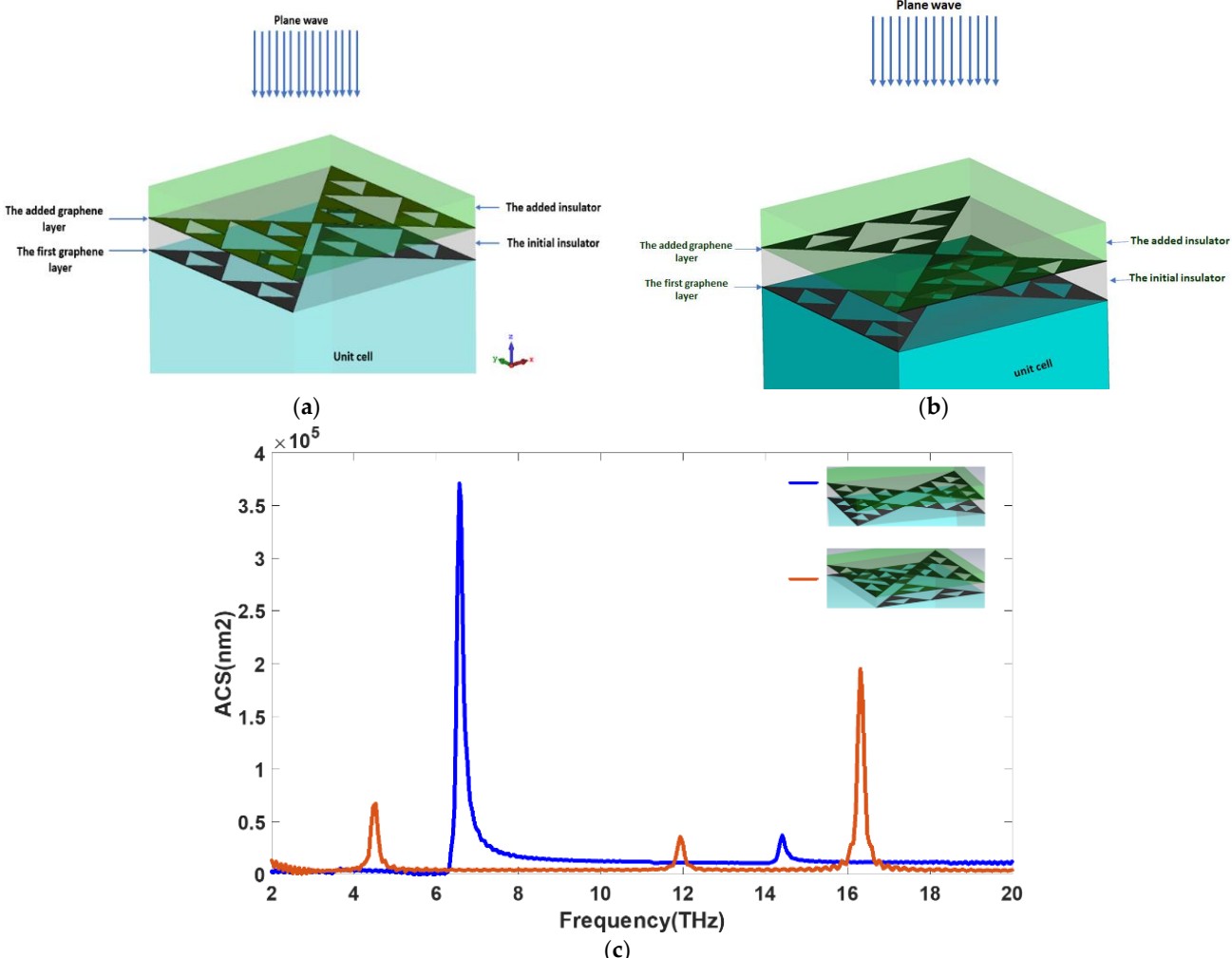

**Figure 4.** (**a**) Double fractal graphene layers embedded in the structure with the similar-state positioning; (**b**) Double fractal graphene layers embedded in the structure with the cross-state positioning; (**c**) ACS spectra of double fractal graphene layers inserted in the two states of the perfect absorber.

**Table 3.** Absorption characteristics of the two-stage absorber in the same-state and cross-state.

| Absorbers with Double Fractal Graphene Layers | Plasmonic Frequency (THz) | The Amplitude of Absorption Peak ($\times 10^5$ nm$^2$) |
|---|---|---|
| In the same-state positioning | 6.5962 | 3.7326 |
| In the cross-state positioning | 16.285 | 1.87564 |
| | 4.52 | 0.66911 |

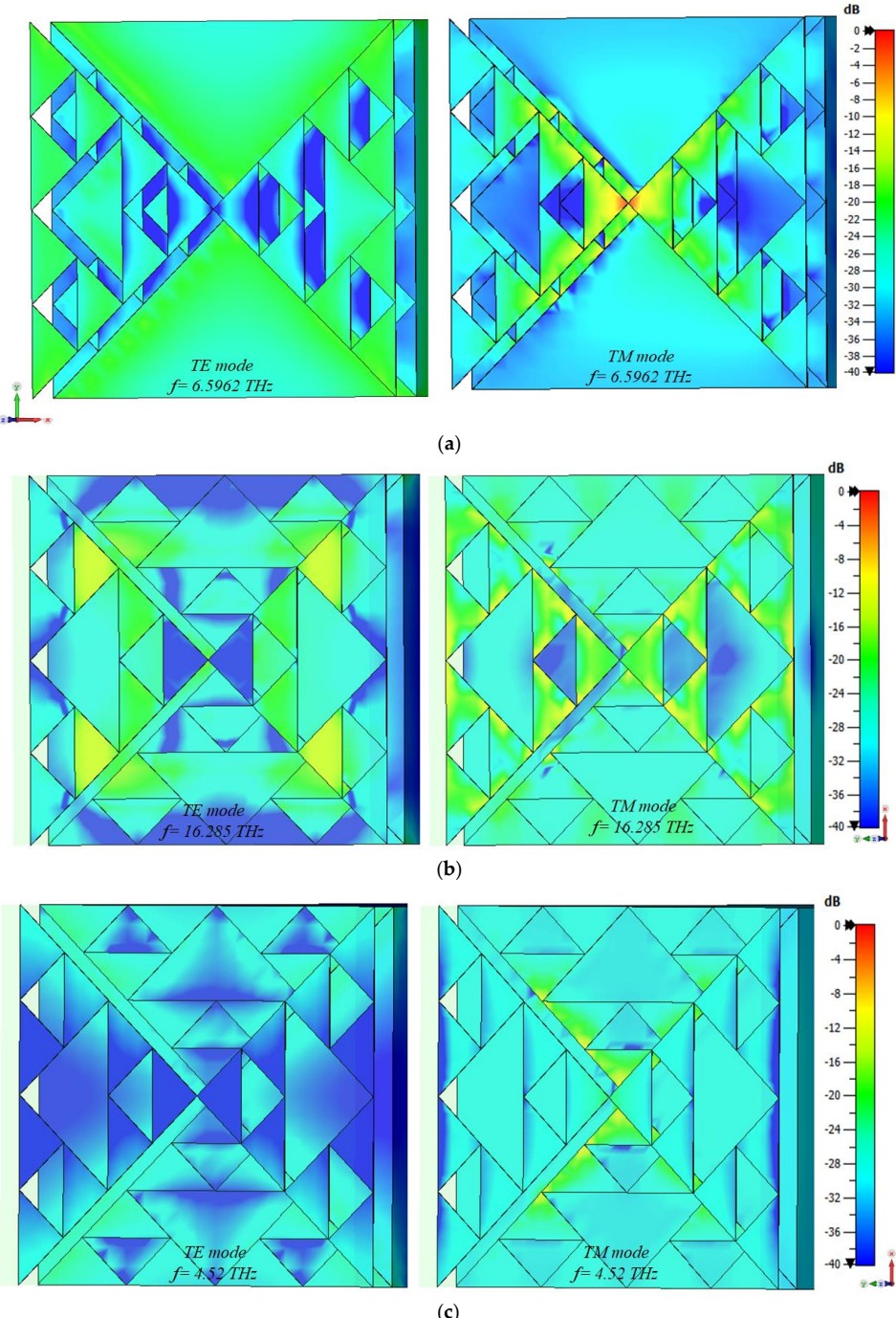

**Figure 5.** The electromagnetic fields distribution for TE and TM modes in plasmonic resonant frequencies of (**a**) the same−state and (**b**,**c**) cross−state configurations.

Now, we intend to focus on the variation of Fermi potentials related to the lower and upper graphene sheets of the structure, placing them on each other, such as in Figure 4a, in order to find the change of the plasmonic mode upon the operating range of the developed absorber. First, the lower graphene layer's Fermi energy is manipulated from 0.4 to 1.5 eV while the Fermi level of the upper graphene layer is fixed at 0.7 eV. In the realistic state, it is impossible to apply a Fermi level of 1.5 eV to this configuration; however, the objective is to present more valuable insights from the variation of the absorption response behavior of the proposed structure on the absorption band. The simulated results in Figure 6a indicate that a blueshift occurs when increasing the Fermi energy. As such, the ACS response experiences a dramatic decline in the absorption peak after a Fermi level of 0.8 eV; in return,

the absorption response switches from the spectral narrowing to a widening state. From the perspective of electronic physics, the Fermi level lies in the forbidden gap between the valence and conduction bands. Thus, increasing the Fermi energy moves the electrons from the valance band to the conduction band. In this situation, the Fermi potential increases and becomes close to the conduction band in terms of energy levels, resulting in a plasmonic modes shift [47]. This means that the variation of the Fermi level displaces the lower graphene sheet's plasmonic mode, changing the resultant plasmonic resonance derived from the interaction of the two graphene layer's plasmonic modes. This phenomenon creates a potential for the spectral narrowing and broadening of the absorption response associated with reducing the absorption amplitude in order to follow an upward trend after a significant fall in the specific operating range of these kinds of fractal absorbers. In the following, the spectral response of ACS witnesses a sharp rise beyond 1.2 eV of the Fermi energy due to the constructive interaction of the plasmonic modes generated by the lower and upper graphene layers embedded in the structure, as illustrated in Figure 6a. Then, by considering the Fermi potential of the lower graphene layer for as much as 0.7 eV, the Fermi level of the upper graphene layer is changed to between 0.4 and 1.5 eV. Similarly, the simulated results in Figure 6b demonstrate a narrowing/widening effect, in which the absorption peak decreases under the Fermi potentials between 0.8 and 1.3 eV upon the absorption band, which can be attributed to the interaction of plasmonic modes available in the structure, as mentioned earlier. Tables 4 and 5 present the distinctions clearly, and we can see quantitive changes of the absorption beamwidth and can compare the results under variations of the Fermi level. Besides, setting the Fermi energy of graphene sheets to 0.9 eV gives the possibility for a spectral broadening of up to 5-fold under electrical manipulation, which then trends to spectral narrowing.

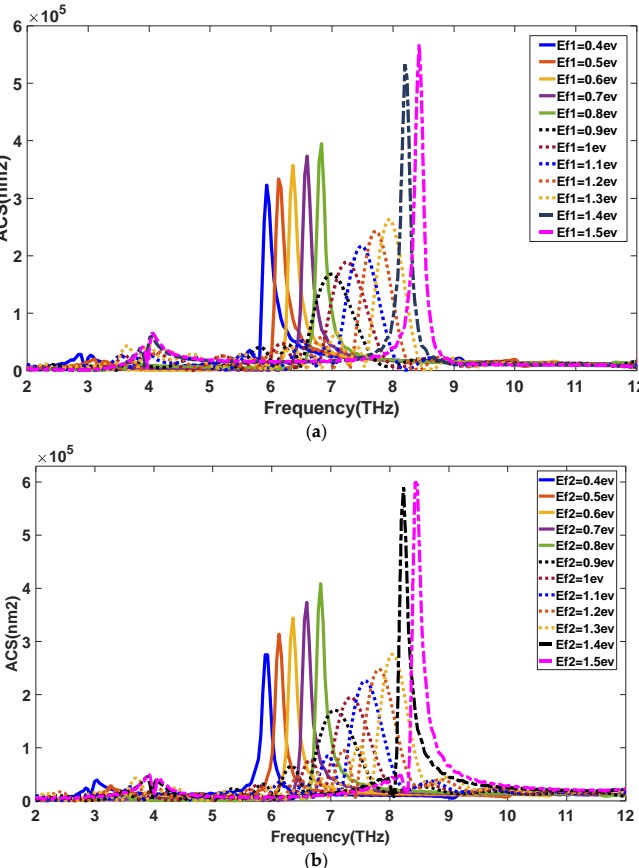

**Figure 6.** (**a**) The variations of ACS spectrum under the lower fractal layer's Fermi level change, (**b**) the variations of ACS under the upper fractal layer's Fermi level change.

**Table 4.** The achieved quantities related to absorption parameters under the change of Fermi energy of lower graphene sheet.

| Fermi Potential (eV) | 0.4 | 0.5 | 0.6 | 0.7 | 0.8 | 0.9 | 1 | 1.1 | 1.2 |
|---|---|---|---|---|---|---|---|---|---|
| Plasmonic Frequency (THz) | 5.9319 | 6.12725 | 6.36172 | 6.5962 | 6.83066 | 6.987 | 7.26052 | 7.495 | 7.7295 |
| The Amplitude of Absorption Peak ($\times 10^5$ nm$^2$) | 3.2249 | 3.13688 | 3.57054 | 3.7326 | 3.94542 | 1.63982 | 1.8853 | 2.16426 | 2.42533 |
| Beamwidth of Absorption Response (GHz) \| (Approximately) | 120 | 120.1 | 123.9 | 116.2 | 116 | 540.3 | 482.2 | 427.8 | 397.3 |

**Table 5.** The achieved quantities related to absorption parameters under the change of Fermi energy of upper graphene sheet.

| Fermi Potential (eV) | 0.4 | 0.5 | 0.6 | 0.7 | 0.8 | 0.9 | 1 | 1.1 | 1.2 |
|---|---|---|---|---|---|---|---|---|---|
| Plasmonic Frequency (THz) | 5.893 | 6.1237 | 6.3617 | 6.5962 | 6.8307 | 7.0651 | 7.3387 | 7.5731 | 7.8076 |
| The Amplitude of Absorption Peak ($\times 10^5$ nm$^2$) | 2.74937 | 3.33415 | 3.44014 | 3.7326 | 4.08488 | 1.70679 | 1.94734 | 2.25912 | 2.47892 |
| Beamwidth of Absorption Response (GHz) (Approximately) | 125 | 118.85 | 116.1 | 116.2 | 114.9 | 542.98 | 498.5 | 440 | 415.93 |

In the continuation of the above framework, the Fermi potentials of the graphene layers embedded in the insulators are simultaneously changed. The simulated results in Figure 7 indicate that a state of spectral narrowing to widening occurs in the absorption response under the simultaneous variation of the graphene layers' Fermi potential when they are set to over 0.7 eV. In addition, the ACS spectrum of the two-stage structure experiences a surge of the plasmonic peak under the constructive effect, stemming from the plasmonic mode's interaction when setting the Fermi level to 1 eV. Interestingly, this optical behavior can be observed again over the absorption band after adjusting the Fermi level to 1.4 eV; however, it is impossible to realize this value of Fermi potential for this size. As a result of the proposed fractal absorbers, it is possible to develop a perfect fractal absorber with the capability of spectral narrowing/widening in its absorption characteristics and adjusting its absorption peak intensity by sweeping the Fermi energy. Besides, the procedure above can be performed for the cross-state of stacked graphene layers embedded into insulators in order to obtain similar results. Table 6 expresses the changes of the absorption parameters under the simultaneous variation of the Fermi potentials related to both fractal graphene layers. Focusing on the extracted data in Table 6 indicates a spectral widening of around 5 folds at 0.8 eV.

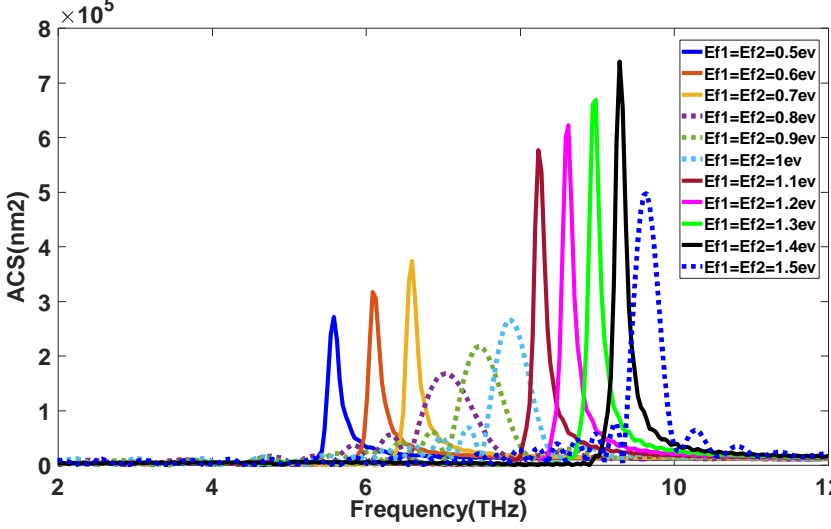

**Figure 7.** The variations of ACS spectrum under the simultaneous change of Fermi energy in both fractal graphene layers.

**Table 6.** The achieved quantities related to absorption parameters under the simultaneous variation of Fermi energy of both graphene sheets.

| Fermi Potential (eV) | 0.5 | 0.6 | 0.7 | 0.8 | 0.9 | 1 | 1.1 | 1.2 | 1.3 |
|---|---|---|---|---|---|---|---|---|---|
| Plasmonic Frequency (THz) | 5.5802 | 6.0882 | 6.5962 | 7.0261 | 7.4559 | 7.8858 | 8.2375 | 8.6283 | 8.98 |
| The Amplitude of Absorption Peak ($\times 10^5$ nm$^2$) | 2.70932 | 3.16794 | 3.7326 | 1.67307 | 2.16996 | 2.65677 | 5.76659 | 6.22226 | 6.68774 |
| Bandwidth of Absorption Response (GHz) (Approximately) | 119.13 | 121.61 | 116.2 | 537.6 | 441.29 | 381.9 | 115.71 | 117.28 | 118.83 |

Afterward, we focus on the multilayer structures composed of graphene layers and dielectric substrates. Insulators are stacked on top of each other in the proposed configuration. The graphene sheets are embedded between them, such as in Figure 8a, which shows a five-stage architecture of fractal absorbers. In addition, the required optical properties of the multi-stage configuration are selected, similar to the initial model used for the primary unit cell with a Fermi level of 0.7 eV. The absorption spectra obtained in Figure 8c show the effect of adding graphene layers placed between insulators' substrates. It is worth noting that graphene layers in different stages form a stronger absorbance intensity in the plasmonic resonance due to the same sheets' admittance matching. Further, an approximate spectral broadening of up to four folds in the absorption band is well-observed in three- and four-stage configurations, as shown in Table 7. It is expected that the increasing trend of the plasmonic amplitude and spectral narrowing/broadening appear by means of adding other stages. The most significant advantage of the proposed structures compared to the recently presented works, such as [27–30], is that the current configurations have the potential to function as a dual switchable absorber in terms of spectral narrowing/widening, which means that the absorption spectrum can widen and narrow by only changing the applied external DC voltage, without any need for geometric manipulation.

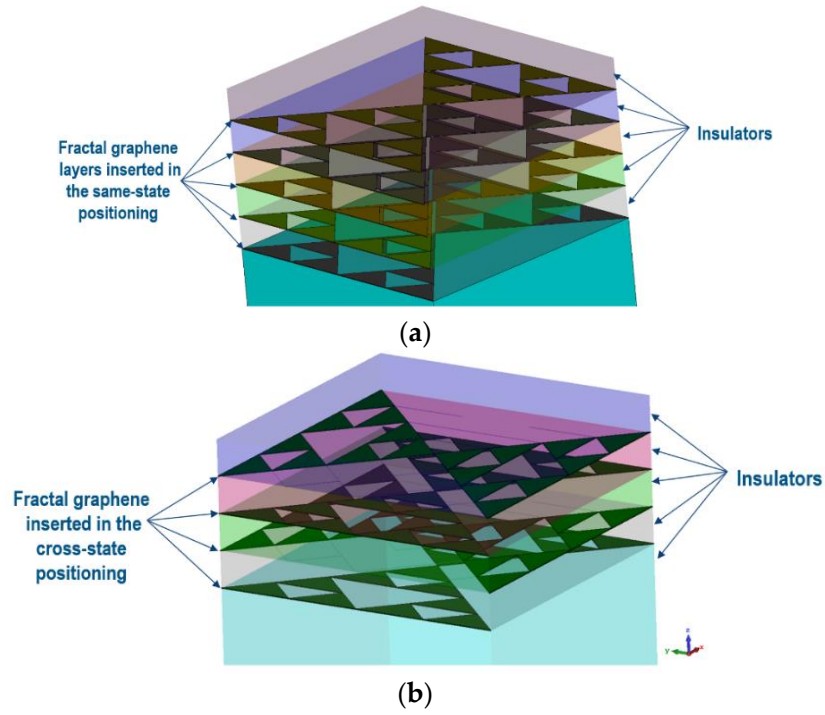

(a)

(b)

**Figure 8.** *Cont.*

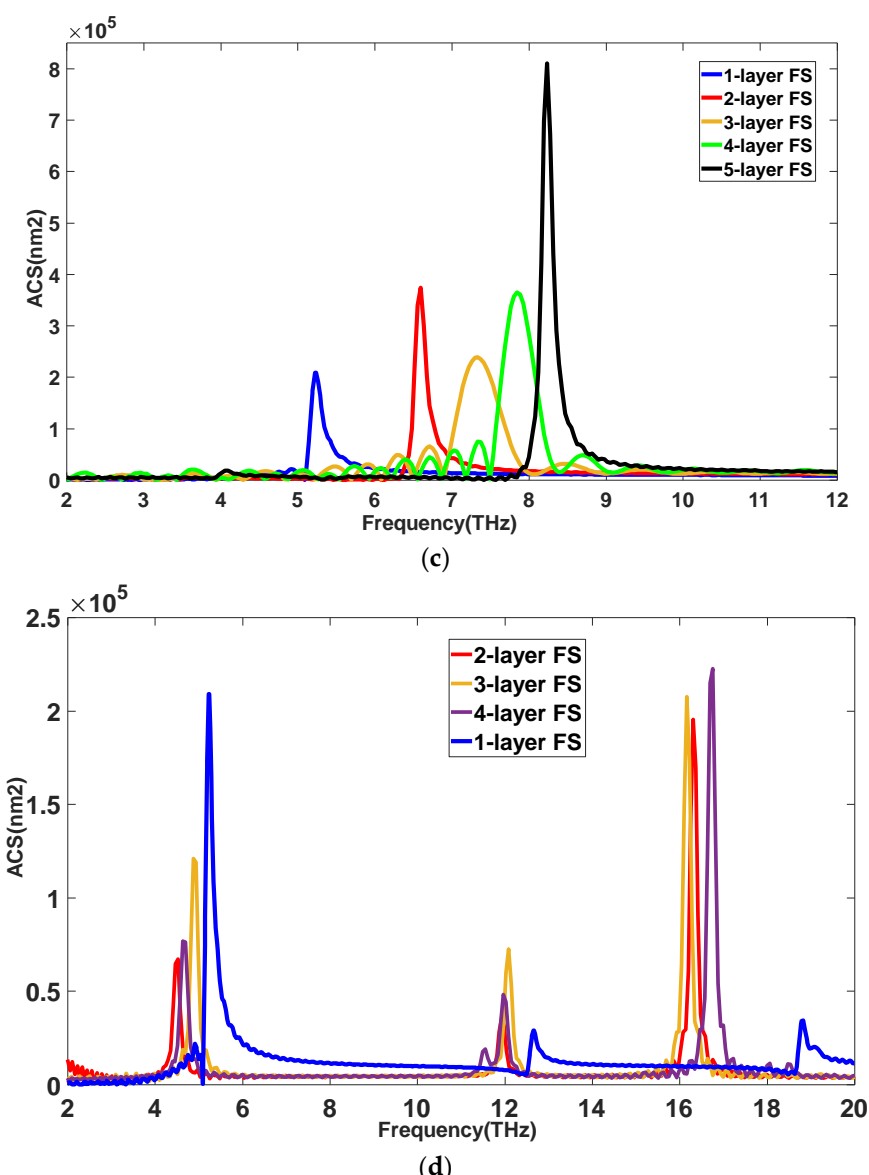

**Figure 8.** (**a**) A schematic of multi-stage absorber structure with the fractal graphene layers inserted in dielectrics at the same-state positioning; (**b**) a schematic of multi-stage absorber structure with the fractal graphene layers inserted in dielectrics at the cross-state positioning; (**c**) ACS spectra achieved as an effect of adding fractal graphene layers at the same-state positioning; (**d**) ACS spectra achieved as an effect of adding fractal graphene layers at the cross-state.

**Table 7.** The achieved quantities related to absorption parameters for multilayer configuration in the similar positioning.

| Number of Graphene Layers Embedded | Single-Layer | Double-Layer | Three Layers | Four Layers | Five Layers |
|---|---|---|---|---|---|
| Plasmonic Frequency (THz) | 5.2285 | 6.5962 | 7.3387 | 7.84669 | 8.2375 |
| The Amplitude of Absorption Peak ($\times 10^5$ nm$^2$) | 2.0892 | 3.7326 | 2.38817 | 3.64856 | 8.09279 |
| Bandwidth of Absorption Response (GHz) (Approximately) | 123.9 | 116.2 | 460 | 360 | 110 |

In another approach, in order to investigate the impact of cross-state positioning for graphene layers in the structure on the absorption response, the graphene sheets are embedded in the cross-state situation, as shown in Figure 8b, one-by-one in succession. The

simulated results in Figure 8d illustrate that the plasmonic resonant peaks are displaced in the absorption band. As such, the amplification of the resonant peaks in the achieved spectra with broadening is well highlighted, whereas broadening due to admittance mismatching arising from the cross-state positioning of the graphene layers occurs less than the previous state positioning. Thus, the predominant plasmonic mode in the absorption band related to two, three, and four-stage configurations of fractal absorbers in the cross-state positioning is moved from a range of 5–6 THz toward a range of 16–18 THz. This denotes the critical role of positioning graphene layers inserted in insulators relative to each other, displacing plasmonic resonances. Table 8 quantitively depicts absorption parameters for multilayer absorbers in the cross-state configuration, which demonstrates that this kind of structure has the potential to function as fractal multiband perfect absorbers, as seen in Figure 8d. For the implementation of the proposed structures, it is clear that the standard microelectronic fabrication planar method can be used. Optical lithography at a submicron level and electron beam lithography at a nanoscale or deep submicron level can be used to image the fractal structure in the proposed perfect absorber. Focusing on the size of the triangles on the fractal graphene layers, which are approximately 100 nm, creating these kinds of sheets is feasible using the common techniques based on photolithography [48] with a suitable resolution. First, the insulator substrate is prepared, and then a graphene sheet can be fabricated by large-scale transfer techniques, and so the photoresist is spin-coated on the graphene sheet. After standard steps in photolithography, the mask is prepared for the desired structure, and, by photoimaging, the pattern can be formed on the resist. After removing the resist on the graphene layer and developing a suitable solvent, the triangle-shaped fractal graphene patterns appear on the top of the substrate. Then, the dielectric layer can be deposited on the graphene layer for passivation.

**Table 8.** The achieved quantities related to absorption parameters for multilayer configuration in the cross-state positioning.

| Number of Graphene Layers Embedded | Single-Layer | Double-Layer | Three Layers | Four Layers |
|---|---|---|---|---|
| Plasmonic Frequency (THz) | 5.2285 | 4.5251<br>16.3086 | 4.875<br>12.0872<br>16.1663 | 4.63828<br>11.9689<br>16.7575 |
| The Amplitude of Absorption Peak ($\times 10^5$ nm$^2$) | 2.0892 | 0.6711<br>1.953 | 1.2086<br>0.72523<br>2.07626 | 3.64856<br>0.48141<br>0.76625 |
| The Bandwidth of Absorption Response (GHz) (Approximately) | 123.9 | 142<br>134 | 157<br>140<br>130 | 140<br>140<br>150 |

In attempting to explore more details concerning the positioning effect of graphene layers over the absorption parameter, the variation of the graphene sheet's positioning state on the insulator is studied. In order to do so, the graphene layer of the initially developed structure is rotated. For this purpose, the unit cell's dimensions are enlarged to as much as 100 nm in order to enable a slight rotation and maintain the absorber's configuration. Then, the graphene layer between the insulators under the angle θ is gradually rotated with the angles 0.5°, 1°, and 1.5° in the structure, as shown in Figure 9a. The extracted results in Figure 9b indicate the ACS's peak amplitude displacement under the rotation of the graphene layer. In fact, a slight variation of the graphene layer's position alters the polarization of the reflected wave undertaking different generated electromagnetic modes, shifting the plasmonic resonance in the absorption band. Furthermore, it is clear that a blueshift also occurs during the rotation of the graphene sheet, following an upward trend in the plasmonic amplitude. Table 9 illustrates the changes in absorption characteristics numerically under the angle rotation of θ.

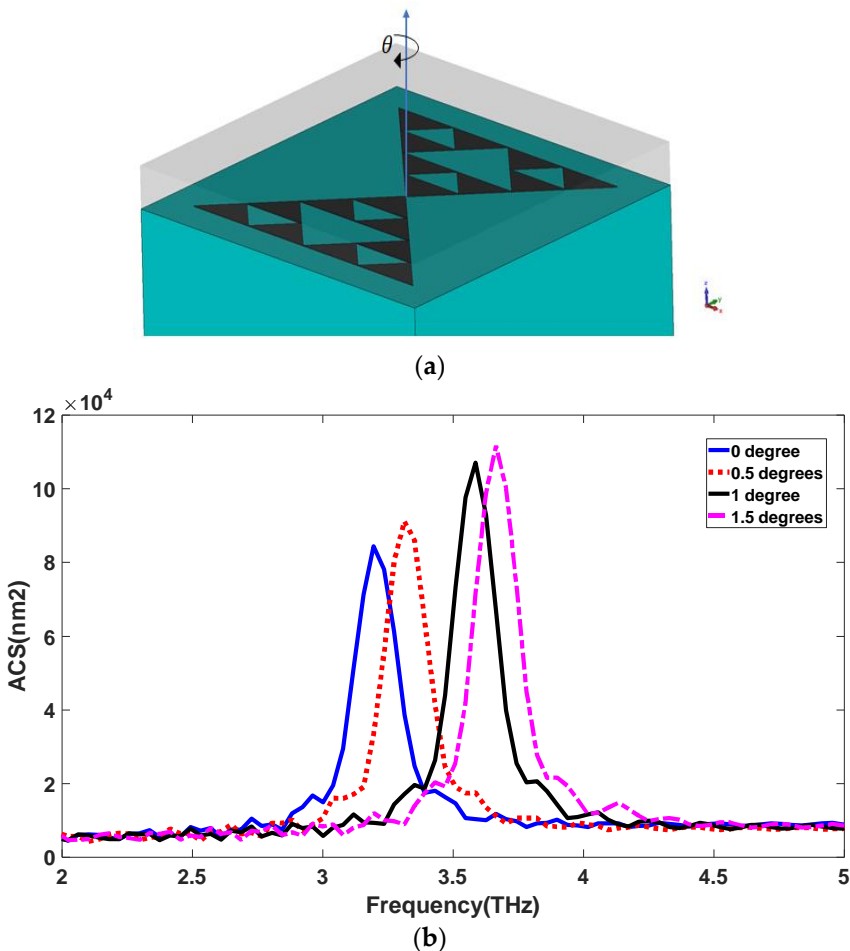

**Figure 9.** (**a**) A schematic of monolayer fractal graphene structure under angular rotation of graphene sheet; (**b**) the variation of ACS spectra achieved under changing angular rotation.

**Table 9.** The variations of absorption characteristics of monolayer fractal graphene structure under the impact of graphene sheet rotation.

| Angular Rotation of Graphene Sheet (°) | 0 | 0.5 | 1 | 1.5 |
|---|---|---|---|---|
| Plasmonic Frequency (THz) | 3.19639 | 3.31363 | 3.58717 | 3.66533 |
| The Amplitude of Absorption Peak ($\times 10^4$ nm$^2$) | 8.43154 | 9.11228 | 10.35872 | 11.16 |

Subsequently, this framework is performed for a two-stage structure with the same configuration of graphene layers, such as in Figure 4a, while the unit cell is 100 nm bigger than it in order to rotate the graphene sheets more easily. First, the lower graphene sheet is rotated with the angles 2.5°, 5°, 7.5°, and 10° when there is no change in the upper graphene layer's position. These manipulations change the modal electromagnetic distribution of the configuration, concluding the variation of the reflected wave's polarization that results in the shifting in the absorption characteristic of the structure, as illustrated in Figure 10a. In other words, due to the interference of modal electromagnetic distribution, which may be constructive or destructive, the plasmonic resonance frequencies and ACS's peak amplitude displace over the absorption band under an increasing angle θ. The comparison of ACS spectra related to θ = 7.5° and θ = 10° demonstrates the influential role of the destructive and constructive interference of modal electromagnetic components. Then, the upper graphene layer is turned under θ = 2.5°, 5°, 7.5° and 10° while the lower graphene is fixed. Figure 10b illustrates the rotation impact of an upper graphene layer on the spectral response of ACS. There is a slight difference between the simulated

results available in Figure 10a,b derived from the electromagnetic field distribution profile variation under changing the admittance matching of configurations relative to each other. Tables 10 and 11 present the difference between the achieved absorption parameters under the rotation of graphene layers separately. Figure 10c also shows the variation of spectral responses of ACS when the rotation of the graphene sheets occurs simultaneously. Table 12 illustrates the absorption specifications of the designed fractal absorber under the simultaneous rotation of both embedded graphene layers.

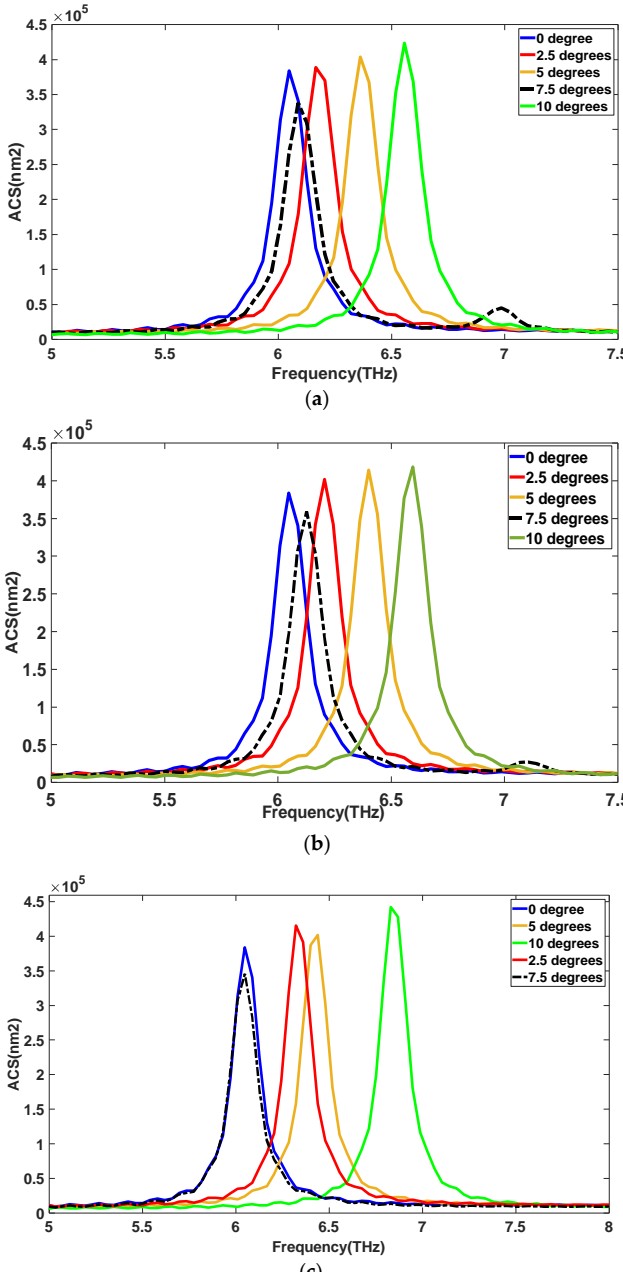

**Figure 10.** (**a**) The variation of ACS spectra achieved by changing the angular rotation of lower graphene sheet in a double-layer structure at the same-state positioning; (**b**) the variation of ACS spectra achieved by changing the angular rotation of upper graphene layer in a double-layer structure at the same-state positioning; (**c**) the variation of ACS spectra achieved by a simultaneous change of the angular rotation of lower and upper graphene sheets in a double-layer structure at the same-state positioning.

**Table 10.** The variations of absorption characteristics of double fractal graphene structure under the rotation of lower graphene sheet.

| Angular Rotation of Graphene Sheet (°) | 0 | 2.5 | 5 | 7.5 | 10 |
|---|---|---|---|---|---|
| Plasmonic Frequency (THz) | 6.0491 | 6.16633 | 6.36172 | 6.08818 | 6.55711 |
| The Amplitude of Absorption Peak ($\times 10^4$ nm$^2$) | 3.83525 | 3.88302 | 4.03278 | 3.37791 | 4.23259 |

**Table 11.** The variations of absorption characteristics of double fractal graphene structure under the rotation of upper graphene sheet.

| Angular Rotation of Graphene Sheet (°) | 0 | 2.5 | 5 | 7.5 | 10 |
|---|---|---|---|---|---|
| Plasmonic Frequency (THz) | 6.0491 | 6.20541 | 6.4008 | 6.12725 | 6.59619 |
| The Amplitude of Absorption Peak ($\times 10^4$ nm$^2$) | 3.83525 | 4.01656 | 4.13524 | 3.58740 | 4.18106 |

**Table 12.** The variations of absorption characteristics of double fractal graphene structure under the simultaneous rotation of graphene layers.

| Angular Rotation of Graphene Sheet (°) | 0 | 2.5 | 5 | 7.5 | 10 |
|---|---|---|---|---|---|
| Plasmonic Frequency (THz) | 6.0491 | 6.3226 | 6.43988 | 6.0491 | 6.83066 |
| The Amplitude of Absorption Peak ($\times 10^4$ nm$^2$) | 3.83525 | 4.15166 | 4.01461 | 3.45296 | 4.4154 |

## 5. Conclusions

In this paper, a framework is presented to design the tunable perfect terahertz absorbers constructed of the fractal triangle-shaped graphene layers between dielectric substrates to have the potential for narrowing and widening the absorption spectrum using a Fermi level change through applying external DC voltage. We design and characterize a perfect absorber's absorption, configured of a single fractal triangle-shaped layer of graphene. The results indicate that the absorbance coefficient is altered by the geometric variations of the graphene layer embedded into insulators. It is also shown that the absorbance level of a fractal absorber developed by a double graphene layer at a similar-state insertion, compared to a monolayer graphene structure, possesses a significant amplification of both the absorption intensity and the shifting of the plasmonic resonance on the absorption band. It denotes that embedding fractal graphene layers into the cross-state positioning configuration provides two separate plasmonic modes with fewer absorption amplitudes than the same-state insertion, due to the admittance mismatching of the graphene sheets. The designed absorber has potential tunability for narrowing and broadening the absorption response and the peak of absorbance rate by changing the Fermi energy using the bias DC voltage. This work indicates that stacking the fractal graphene layers between similar dielectric substrates increases the absorption bandwidth remarkably and manipulates the resonant peak's amplitude, which can be adjusted by setting the Fermi levels of different fractal graphene layers. In addition, multilayer structures in the cross-state give the possibility for the creation of multiband absorbers that can be interchanged to different absorption frequencies. As such, a slight rotation of the fractal graphene layer embedded into dielectric substrates can move the plasmonic mode over the absorption band, employing switchable absorbers. The presented mechanism indicates the potential for tunability, spectral narrowing/broadening, and plasmonic resonance switching by configuring multi-stage structures that can promote these kinds of fractal absorbers as a promising candidate for applications in sensory photonic devices and optical integrated circuits.

**Author Contributions:** A.M. designed the structures and simulations and wrote the paper. A.R. designed the concept, supervised the work, edited the paper, and reviewed it, and I.B. supervised the work, technically investigated, and completed editing and reviewed it. N.G. had a final check and revised it in terms of writing style. All authors have read and agreed to the published version of the manuscript.

**Funding:** This work was supported by the European Research Consortium for Informatics and Mathematics (ERCIM), and Research Council of Norway funded project CIRCLE (287112).

**Institutional Review Board Statement:** Not applicable.

**Informed Consent Statement:** Not applicable.

**Data Availability Statement:** Not applicable.

**Conflicts of Interest:** The authors declare no conflict of interest.

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
