# Peer review of "Design and Simulation of Terahertz Perfect Absorber with Tunable Absorption Characteristic Using Fractal-Shaped Graphene Layers"

_photonics, doi:10.3390/photonics8090375_

Round 1

Reviewer 1 Report

1. What is the defination of the absorption cross-section (ACS)? Usually, the absorption is no more than 100%, but the y-axis in Fig. 2a shows the ACS with the unit of nm2, which are not well explained in the text. Please clarify this.
2. Following the above question, as the frequency shifts, the magnitude of ACS will dramatically be changed. What will it affect for the absorption?
3. Some simple structures based on graphene can also be used for the design of multi-band absorbers, such as the following reference [R1]. Compared with some recently reported works, what is the advantage of your work? Observed from the proposed structure in the manuscript, it seems the absorber is very complicated using fractal structure but the absorption performance is moderate. Please cite some recently reported works including [R1] and more importantly address your own advantages over these reported works. 
[R1]“Tunable multi-band terahertz absorber using single-layer square graphene ring structure with T-shaped graphene strips,” Optics Express, 28(8), 11482-11492, 2020. 
4. Do the authors consider how to fabricate the proposed absorber with fractal structure? In my opinion, it is very hard to pattern the structure in Fig. 1(b). Please clarify this.

Author Response

Dear Reviewer,

I attached a revision letter to this email regarding your comments. Please find it.

Thank you very much for dedicating your time to this work.

My best regards

Reviewer 2 Report

I have carefully read the manuscript entitled: "Design and Simulation of Terahertz Perfect Absorber with Tunable Absorption Characteristic Using Fractal-Shaped Graphene Layers" by Maghoul et al. The field of perfect absorbers at THz frequencies is an active field with constant interest. The specific manuscript is well written, the statements are adequately supported by the presented results and the quality of the current state of the manuscript is suitable to be published in Photonics. My suggestion is to be accepted for publication in the current form. 

My main question to the authors is, how feasible is the fabrication of such structures in graphene so the proposed design could be experimentaly realized?

Author Response

(The authors gave the same response as above.)

Reviewer 3 Report

In manuscript, the authors presented numerical calculations for design  Terahertz perfect absorber with fractal shaped graphene layers. Although the authors investigated many parameters which indeed change the absorption peak and width,  the deep analysis or physics reasoning, I think, is missing yet.  To support the conclusion, the authors need to improve further the manuscript. I would thus suggest a big modification.  My comments are here:

  • The definition of “perfect absorber” .   The calculated ACS changes with different setting, varying in big range. To which extent, we can call it “perfect absorber”?  The relation between the absorption efficiency and the ACS is also need to be stated.
  • Effect of the insulator h2 in Fig.1 on absorption.  I understand that the Gold acts as a mirror to enhance the absorption. What is the effect of the insulator, e.g. the insulator height, refractive index? Why the Line 336 introduce another semiconductor  substrates? The insulator parameter not only  affect the absorption of  a single layer, but also  strongly affect the coupling between the stacked layers. I would say it is a very important parameter to evaluate the absorption of the graphene layer. The related physics mechanism is also needed.
  • The physics mechanisms  for tuning the Fermi energy to board the resonant width. The resonant width in Fig.6, Fig.7, Fig.8c, experiencing narrowing, boarding and narrowing again, which is indeed very interesting. Here, deep analysis is desirable. To understand this,  ACS of a single layer  for Ef up to 1.5 eV is also needed.

In summary, the manuscript, to some extent, is interesting and novel. But a comprehensive analysis is needed to get a clear picture for understanding and tailoring the perfect absorption of the graphene.

Author Response

Dear Reviewer,

Thank you

Round 2

Reviewer 1 Report

The authors have addressed my concerns.

Author Response

Dear Reviewer,

My best regards

Reviewer 3 Report

Thank the authors reply which solve part of my comments.  But I am not very satisfied with one question about the absorption broading.  With the addition of single layer along with changing the Fermi enerngy, Fig.3 shows the broading at Ef= 1.5eV. Then I would suggest to explain that: why in Fig. 6 or Fig.7 (similar to Fig.8) such broading comes with reducing Ef =0.9 eV, and why the absorption bandwidth narrows again at larger Ef.  Resonable explanations is needed. 

Author Response

Dear Reviewer,

My best regards
